# SELECTIVE PERCEPTION: LEARNING CONCISE STATE DESCRIPTIONS FOR LANGUAGE MODEL ACTORS

## ABSTRACT

It is increasingly common for large language models (LLMs) to be applied as actors in sequential decision making problems in embodied domains such as robotics and games, due to their general world knowledge and planning abilities. However, LLMs are not natively trained for embodied decision making problems, and expressing complex state spaces in text is non-trivial. Exhaustively describing high-dimensional states leads to prohibitive inference costs and impaired task performance due to distracting or irrelevant information. Previous LLM actors avoid the issue by relying on *hand-engineered*, *task-specific* protocols to determine which features to communicate about a state and which to leave out. In this work, we propose BLINDER (Brief Language INputs for DEcision-making Responses), a method for learning to select concise and helpful sets of state features for LLM actors. BLINDER learns a value function for task-conditioned state descriptions that approximates the likelihood that a state description will result in optimal actions. We evaluate BLINDER on the challenging video game NetHack and a real-world robotic manipulation task. Our method improves task success rate by 77% and 14% on NetHack and robotic manipulation respectively, reduces model input length by 83%, and generalizes well to LLM actors of various size and quality.

## 1 INTRODUCTION

Large Language Models (LLMs) excel at a variety of tasks and have been shown to be skilled actors for decision making in games and robotic planning (Ichter et al., 2022; Huang et al., 2022b; Liang et al., 2022; Singh et al., 2022; Skreta et al., 2023; Zhao et al., 2023; Lin et al., 2023; Wake et al., 2023). The input to an LLM actor typically consists of a description of the current task and state, and the actor then outputs actions to accomplish the task. For example, consider a robot tasked with arranging items on a table according to a task specification (Valmeekam et al., 2022). An LLM actor outputs high-level actions that the robot can take to manipulate the items such as "move the apple to coordinate X". While it would be simple to communicate the state of the items using a camera sensor, most LLM actors require exclusively language-based inputs. Previous work has side-stepped this problem by either (1) exhaustively describing the state (Valmeekam et al., 2022) or (2) engineering an ideal input format for the current task (Huang et al., 2022b).

A naive approach to language grounding would be to provide an exhaustive set of textual state features that describes everything about the state (Valmeekam et al., 2022), independent of the current task. While an exhaustive task-agnostic set of state features can be readily fed into an LLM actor, such inputs can be excessively long and contain potentially distracting features in the state description. Full state descriptions can impede performance and increase inference costs by including unnecessary state information.

Alternatively, manually engineered state descriptions for an LLM actor can increase performance and decrease the length and cost of inputs (Huang et al., 2022b; Liang et al., 2022; Singh et al., 2022). However, manual descriptions are typically also task-specific. This limits a system's ability to generalize to new tasks that may require descriptions of different portions of the state. Additionally, manual state descriptions assume expert task knowledge, further limiting the application of this approach in the real world. For state descriptions to be more task-general, without infeasible expert burden, the system should learn to automatically generate task-conditioned state descriptions.

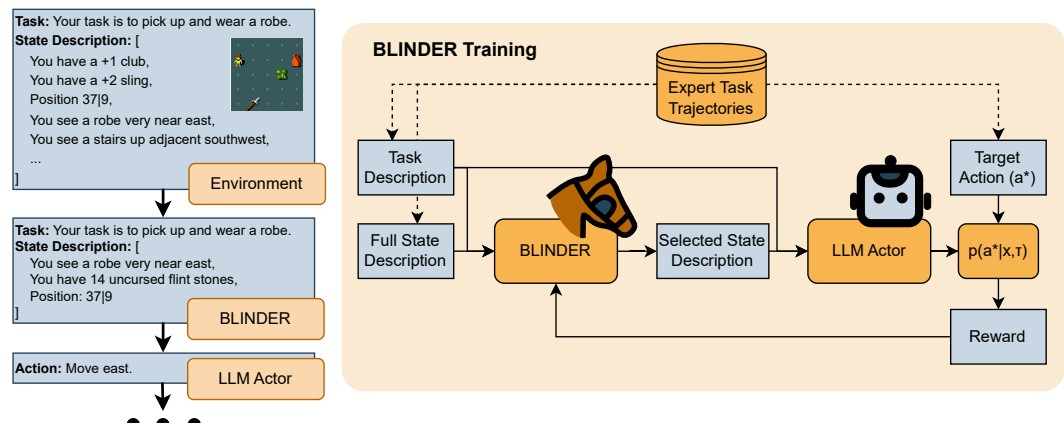

Figure 1: BLINDER selects optimal state descriptions, reducing context length and removing distracting information. BLINDER is trained via RL by learning to maximize the probability of optimal actions from an LLM actor, as shown on the right.

We propose **BLINDER (Brief Language INputs for DEcision-making Responses)**, a method for learning to automatically select task-conditioned state descriptions from a set of task-independent state features. For example, consider a robot tasked with arranging fruit by size. The set of state features may include items other than fruit on the table or attributes of the fruit not relevant to size. BLINDER selects state features relevant to the position and the size of each fruit. By selecting relevant state features for the LLM actor's state description, BLINDER improves performance and decreases input length and compute costs. This reduction in context length can further be applied to environments where there are exponentially many features.

Unlike prior work, BLINDER automatically constructs state descriptions without requiring task-specific prompt engineering. We frame the construction of state descriptions as a reinforcement learning (RL) problem, as shown in Figure 1. Given a small set of expert task demonstrations—including task descriptions, exhaustive state features, and expert actions—and an LLM actor, we assign rewards to proposed state descriptions using the likelihood of expert actions from the actor. We train a value model with these rewards, and, at inference time, we maximize value to select state descriptions for new tasks, unseen during training.

We evaluate BLINDER on challenging NetHack (Küttler et al., 2020) and robotic item arrangement tasks. Notably, all of our evaluation is done on held out test tasks that were not demonstrated during training. By finetuning an LLM to represent our value function, we find that BLINDER generalizes well to tasks that include novel entities and task compositions. We also show that BLINDER's learned state descriptions are intuitive enough to generalize across LLM actors, allowing us to train with one LLM actor and evaluate on another. In our experiments, BLINDER compared with exhaustive state descriptions reduces context lengths by 83%. We also compare with zero-shot summaries of exhaustive state descriptions from an LLM 4x larger than BLINDER's. Our experiments show that BLINDER outperforms both baselines by an average of 77% and 14% for NetHack and robotic tasks respectively. Finally, we compare with hand-engineered state descriptions and show that BLINDER performs competitively despite not relying on the expert knowledge that manual descriptions require.

## 2 RELATED WORK

### 2.1 LANGUAGE FOR STATE DESCRIPTIONS

Language has been viewed as a promising medium for communicating task requirements (Chevalier-Boisvert et al., 2019; Anderson et al., 2018; Ku et al., 2020; Shridhar et al., 2020) or state information (Andreas et al., 2017; Tam et al., 2022; Mu et al., 2022) due to its compositionality and interpretability. Since the prevalence of LLMs for planning and decision making, it has become essential to de-

scribe state features in langauge to better ground LLM plans in the environment state (Huang et al., 2022b). However, grounding is a nontrivial task especially in spatial and real world settings.

## 2.2 FOUNDATIONAL MODELS FOR PLANNING

LLMs have recently become popular for planning and high-level decision making in robotic tasks (Ichter et al., 2022; Huang et al., 2022a; 2023; Vemprala et al., 2023) and other sequential decision making benchmarks (Nottingham et al., 2023; Kim et al., 2023; Liu et al., 2023a; Liang et al., 2023). However, LLMs are not natively grounded in the state of the environment when prompted for planning, resulting in sometimes nonsensical actions. While some learned grounding techniques exist (Ichter et al., 2022), the most straightforward way to ground LLMs in state is to include state descriptions in the LLM context (Huang et al., 2022b). Many LLM applications for robotics achieve this by defining a formal language for planning that includes definitions for state features (Liang et al., 2022; Singh et al., 2022; Skreta et al., 2023; Zhao et al., 2023; Lin et al., 2023; Wake et al., 2023). Utilizing multimodal models—which accomplish grounding by using both visual and textual inputs—may be posited as an alternative approach for planning within foundational models Driess et al. (2023); Guhur et al. (2023). Although multi-modal models offer a promising solution to the grounding problem, current multimodal models tend to be domain-specific Guhur et al. (2023); Jiang et al. (2022) and lack robustness across tasks. Additionally, these models are not as widely available for public use Reed et al. (2022); Driess et al. (2023). Consequently, the use of text-only models, which are easily accessible, continues to be a crucial approach for addressing planning tasks.

## 2.3 LEARNING INPUTS FOR PRETRAINED MODELS

An extensive range of studies investigate efficient techniques for learning effective input for a variety of pretrained language models (Liu et al., 2023b; Qiu et al., 2020) in discrete (Shin et al., 2020; Liu et al., 2023b; Schick & Schütze, 2020; Gao et al., 2021b; Shi et al., 2022) or continuous (Qin & Eisner, 2021; Liu et al., 2021; Lester et al., 2021; Zhong et al., 2021) input space. This line of work is also studied in pretrained vision models (Gao et al., 2021a; Zhou et al., 2022a; Ju et al., 2022; Zhou et al., 2022b; Yao et al., 2021).

Among these prompt tuning methods, Deng et al. (2022) use RL to optimize the prompt for a variety of NLP tasks. Perhaps the closest work to our own is that of Zhang et al. (2023), which introduces an RL agent trained to edit language models input at test time for solving NLP benchmarks in a more sample-efficient way. Their approach differs significantly from our own and focuses on static NLP tasks rather than sequential tasks. However, the prospect of applying similar techniques to modify the state description in sequential tasks could serve as a potential avenue for our future work.

Recent methods have explored the idea of compressing input tokens for shorter context length (Mu et al., 2023; Chevalier et al., 2023), which is related to our effort to learn pruned state descriptions but does not meet the needs of sequential decision making nor maintain intuitive inputs that generalize between models.

## 3 BACKGROUND

It is important to distinguish the two levels of decision-making present when selecting state descriptions for an LLM actor. At one level of decision-making, the LLM actor $LLM : \mathcal{X} \times \mathrm{T} \rightarrow A$ receives a task description string $\tau \in \mathrm{T}$ and state description $x \in \mathcal{X}$ as input, and the LLM actor outputs an environment action $a \in A$ to execute. The set of possible state descriptions $\mathcal{X}$ is the power set of all possible state feature strings $\omega \in \Omega$, $\mathcal{X} = \{x | x \subseteq \Omega\}$. Each linguistic feature $\omega \in \Omega$ is a minimal atomic statement that describes some aspect of the environment's state. For example, a given $\omega$ may indicate the presence of an object in a scene (Huang et al., 2022b), the completion of subgoals (Nottingham et al., 2023), or an attribute of or relationship between environment entities.

Independent from the LLM actor's decision making, BLINDER selects which subset of possible features in $\Omega$ to include in the state description $x$. We frame the selecting of state descriptions as a RL problem with a Markov Decision Process (MDP) in which state descriptions are assembled one feature at a time. In this MDP, the state at step $t$ consists of the state description so far $x_t \in \mathcal{X}$, and the action space includes individual features $\omega_t \in \hat{\Omega}$ to add to $x_t$ and form the updated state

description $x_{t+1}$. Here $\hat{\Omega} \subseteq \Omega$ is the subset of all possible state features that are factually true in the current environment state. This description-building MDP has deterministic transitions where each step adds a new state feature to the state description. We define our transition function $P : \mathcal{X} \times \Omega \rightarrow \mathcal{X}$ as $P(x_t, \omega_t) = x_t \cup \{\omega_t\}$. Our reward function $R : \mathcal{X} \times \mathrm{T} \rightarrow \mathbb{R}$ represents the likelihood that a state description yields an optimal target action from the LLM actor (See Section 4.2). Finally, we learn a policy $\pi : \mathcal{X} \times \mathrm{T} \rightarrow \hat{\Omega}$ that iteratively constructs task-conditioned state descriptions.

## 4 BLINDER

We propose learning to automatically select a state description $x$ from an exhaustive set of factual state features $\hat{\Omega}$ given a task $\tau$. Our goal is to automatically learn optimal state descriptions without requiring hand-engineered descriptions for each task nor datasets of annotated examples of state descriptions. Instead, we suggest a method that learns to select state descriptions for both seen and novel tasks that avoid distracting information and reduce the input length for LLM actors.

### 4.1 SELECTING STATE DESCRIPTIONS

Our method, BLINDER, defines a value-based policy,

$$\pi(x_t, \tau, \hat{\Omega}) = \underset{\omega_t^* \in (\hat{\Omega} - x_t)}{\arg\max} V_\theta(x_t \cup \{\omega_t^*\}, \tau), \quad (1)$$

that selects unused state features $\omega_t^* \in \hat{\Omega} - x_t$ to add to $x_t$. $V_\theta : \mathcal{X} \times \mathrm{T} \rightarrow \mathbb{R}$ is a value function trained to approximate the return of a state description (see Section 4.2). Since the transition model $P(x_t, \omega_t) = x_t \cup \{\omega_t\}$ is known and deterministic, we can use a state value function directly to define our policy.

We use $\pi$ to construct a state description $x$ out of state features $\omega$. Starting with $x_0 = \emptyset$, $\pi$ iteritively adds $\omega_t^*$ to $x_t$ as long as it increases in value and there are still state features to add (see Algorithm 1). Once no state features remain, $\hat{\Omega} - x_t = \emptyset$, or no added state

---

**Algorithm 1** State Description Selection

**Require:** $V_\theta$, $\tau$, $\hat{\Omega}$, $\pi$
  $x_0 \leftarrow \emptyset$
  $t \leftarrow 0$
  **while** $|\hat{\Omega} - x_t| > 0$ **do**
    $\omega_t^* \leftarrow \pi(x_t, \tau, \hat{\Omega})$
    **if** $V_\theta(x_t \cup \{\omega_t^*\}) > V_\theta(x_t)$ **then**
      $x_{t+1} \leftarrow x_t \cup \{\omega_t\}$
      $t \leftarrow t + 1$
    **else**
      break
    **end if**
  **end while**
  $x_f \leftarrow x_t$
  **return** $x_f$

---

feature increases the value of the state description, we terminate and use the last $x_t$ as the final state description, $x_f$. This state description is then used to prompt the LLM actor for the next action in the downstream task,

$$LLM : x_f \times \tau \rightarrow a. \quad (2)$$

### 4.2 STATE DESCRIPTION VALUE

The key to BLINDER is learning a useful and general value function $V_\theta$. We first define the reward function $R$ used to train $V_\theta$.

We propose using action likelihood from an LLM actor to define rewards for BLINDER. We collect a set of expert trajectories composed of state features, task descriptions, and target actions $D_e = \{(\hat{\Omega}_0, \tau_0, a_0^*), (\hat{\Omega}_1, \tau_1, a_1^*), ...\}$ and define the following sparse reward function,

$$R(x_t, \tau) = \begin{cases} \mathbb{E}_{a^* \sim (D_e | x_t \subset \hat{\Omega}, \tau)} \left[ LLM(a^* | x_t, \tau) \right] & \text{if } x_t = x_f \\ 0 & \text{otherwise,} \end{cases} \quad (3)$$

where $a^*$ is sampled from tuples in $D_e$ with matching $\tau$ and where $x_f$ is a valid subset of $\hat{\Omega}$. Intuitively, we reward BLINDER to maximize the likelihood that $x_f$ elicits the same action from the LLM actor as the target action $a^*$ from the expert trajectory.

| Task Description | Manual | Zeroshot | BLINDER |
|---|---|---|---|
| Drink the potion and navigate to the stairs down. | You see a stairs down very near east. | You see a stairs down very near east. You see a lava very near north northeast. | You see a stairs down very near east. You have 20 uncursed flint stones (in quiver pouch). Time: 5. Constitution: 13. |
| Pick up and eat the apple. | You see a apple very near east southeast. | You have an apple. | You see a apple very near east southeast. You have a blessed +2 sling (alternate weapon; not wielded). Time: 1. Condition: None. |

Table 1: Qualitative example output from BLINDER and baselines. Blue text indicates the most relevant information for a task, while red text indicates distracting or inaccurate information for the current task. Note that zeroshot descriptions often contain information that hinders performance. Although BLINDER sometimes includes unnecessary information, any extra included state features do not distract from the current task.

Although $R$ depends on a specific LLM actor, we find in our experiments that BLINDER selects intuitive state features, consistently removing the most distracting information and including the most relevant information as illustrated in Table 1. In contrast, previous work that optimizes inputs for a frozen LLM often learns nonsensical language inputs (Shin et al., 2020). BLINDER generates intuitive state descriptions due, in part, to being constrained to the set of grounded state features. Because of this, we find that BLINDER generalizes well to other LLM actors. In Section 6.1, we demonstrate that we can train BLINDER with a smaller LLM actor and then use it to improve the performance of a larger black-box LLM actor.

### 4.3 LEARNING THE VALUE FUNCTION

To learn $V_\theta$, we choose to finetune a pretrained LLM with a value head to maintain generalizability to new tasks and state features. While we refer to $x$ as a set, the text input to $V_\theta$ is ordered using $x$'s insertion order. Given a dataset of precollected experience $D_V = \{(\tau^0, x_t^0, r_f^0), (\tau^1, x_t^1, r_f^1), ...\}$ where $r_f^i = R(x_f^i, \tau^i)$, BLINDER optimizes $V_\theta$ with the loss:

$$J_V(\theta) = \mathbb{E}_{\tau, x_t, r_f \sim D_V} \left[ \mathcal{L}\left( V_\theta(x_t, \tau), \; r_f \right) + \phi \right] \tag{4}$$

$\mathcal{L}$ is a loss metric for the value (we implement quantile regression), and $\phi$ is a Kullback-Leibler penalty for normalizing $V_\theta$, commonly used when finetuning LLMs with RL (Stiennon et al., 2020; Leblond et al., 2021). In practice, evaluating the reward function $R(x_t, \tau)$ requires running inference on an expensive LLM actor. To lower training costs, we collect $D_V$ only once prior to training using a random policy $\pi_r$. Trading off accuracy for computational efficiency, we train $V_\theta$ to model $V_{\pi_r}$ as a cost-effective approximation of the greedy optimal value function $V_{\pi^*}$. Despite this, BLINDER still learns to produce highly effective and concise state descriptions that improve LLM actor performance and are competitive with manually engineered state descriptions.

## 5 EXPERIMENT SETUP

In each of our experiments, BLINDER is trained on a set of training tasks and is provided five expert trajectories for each task. The trajectories are then used to generate rewards and train the value function for BLINDER.

We use two different LLM actors in our experiments. (1) Our **zeroshot flan-T5 actor** is a three billion parameter `flan-t5-xl` model (Chung et al., 2022). This is the model BLINDER uses for training. For each action, $a$, at each environment step, we calculate the geometric mean of the logits of the tokens in $a$ and use this to sample an action from the LLM actor. (2) our **Fewshot GPT3.5**

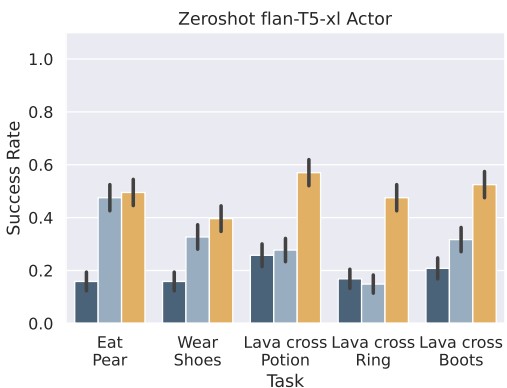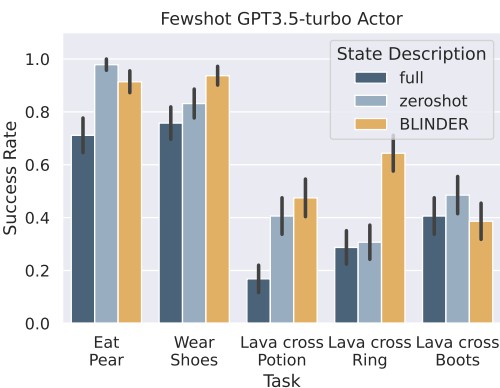

Figure 2: Success rate on NetHack test tasks using zeroshot and fewshot actors. We compare BLINDER against two baselines: (1) using the full state description and (2) instructing a zeroshot LLM to summarize the full state description. Although BLINDER is trained to optimize inputs for the flan-T5 actor, it generalizes well to the much larger GPT3.5 Turbo Actor.

**Turbo actor** uses `gpt3.5-turbo-0301`* to select actions from a list of admissible actions given a task and state description and several fewshot examples. Unless otherwise stated, this model uses six-shot examples. We use this model to evaluate BLINDER's ability to generalize between LLM actors. BLINDER itself finetunes a 780 million parameter `flan-t5-large` model for its value function. See Appendix B for model prompting details.

## 5.1 BASELINES

Previous NLP approaches for learning LLM inputs focus on maximizing performance on a dataset by conditioning on a special set of learned uninterpretable tokens. As a result, we have to define our own baselines for comparison. As an initial baseline, we compare state descriptions from BLINDER with exhaustive state descriptions. We refer to the exhaustive state description that includes all feaures of $\hat{\Omega}$ as the "full" state description in our experiments. We obtain "full" state descriptions $\hat{\Omega}$ for NetHack from the natural language wrapper recommended by Küttler et al. (2020)†. The resulting sets of state features vary in size in the range $30 \leq |\hat{\Omega}| \leq 40$. For our robotic task, we use a list of the pairwise spatial relationship of each object to every other object, totalling 90 state features for our task setup. These "full" states contain mostly irrelevent or distracting information (see Tables 6 and 7 in the appendix). On average, BLINDER removes 83% of the original state features.

We also compare state descriptions from BLINDER to those of a zeroshot `flan-t5-xl` model (Chung et al., 2022) prompted to summarize relevant state features from $\hat{\Omega}$. We refer to these as "zeroshot" state descriptions in our experiments. Note that this model has 4x the number of parameters as the `flan-t5-large` model we use to train BLINDER. Using a `flan-t5-large` model for the zeroshot baseline failed to produce useful descriptions.

Additionally, we compare with hand-engineered state descriptions. To create these, we specify a set of relevant keywords for each task. Then, we include all state features that contain those keywords. We refer to this baseline as "manual" state descriptions in our experiments. Because such summaries have to be manually designed for each task, this baseline is not usually available for test tasks and would not generalize across tasks like BLINDER does.

---

*https://platform.openai.com/docs/models/gpt-3-5
†https://github.com/ngoodger/nle-language-wrapper

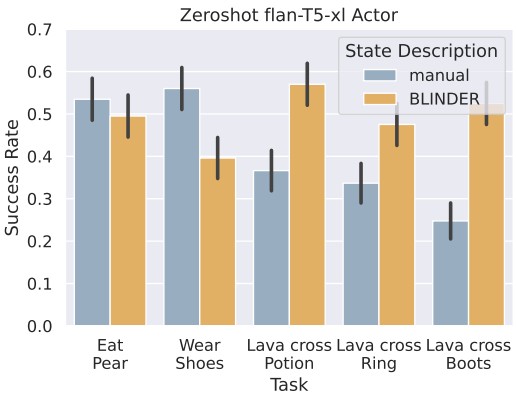
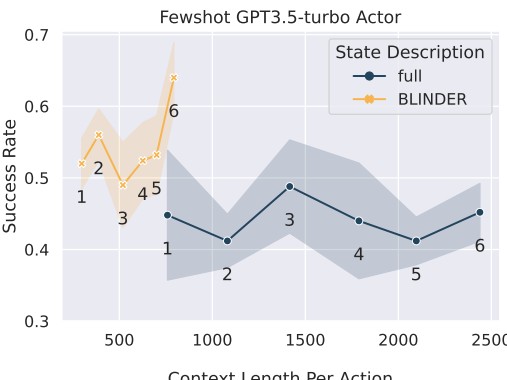

Figure 3: We compare automatic BLIN-DER state descriptions to expert, manually specified state descriptions. BLINDER has competitive performance with manual state descriptions, even outperforming it on the multi-step lava crossing tasks.

Figure 4: Average success rate across all NetHack tasks as a function of context length. Markers indicate the number of few-shot examples provided to the LLM actor, and error bars show standard error across random sets of fewshot examples.

## 6    NETHACK EXPERIMENTS

We evaluate BLINDER on NetHack tasks using the Minihack library (Samvelyan et al., 2021). NetHack, recently proposed as a testbed for AI research (Küttler et al., 2020), is a grid-based dungeon crawler with complex dynamics and large observation and action spaces. We select a set of five training tasks from the Minihack environment zoo: `Room-Monster-5x5`, `Eat`, `Wear`, `LavaCross-Levitate-Potion-Inv`, and `LavaCross-Levitate-Ring-Inv`. We use just 25 expert trajectories for training with a total of 148 pairs of states and expert actions that required under an hour for a human annotator to collect.

We design five difficult custom test tasks to evaluate the performance of BLINDER. Two tasks, `Eat pear` and `Wear shoes`, are variants of the `Eat` and `Wear` training tasks but with different target items, larger rooms, monsters, and distractor items. We also define three `Lava cross` test tasks. Unlike the training variants of this task, the item needed to cross the lava does not start in the player inventory, necessitating improved multi-step planning from the LLM actor. The boots variant of the lava cross task was not seen during training. See Appendix C for an example trajectory.

### 6.1    BLINDER GENERALIZATION

Figure 2 shows success rate on test tasks in the NetHack domain, comparing BLINDER to other baselines that can generalize to unseen tasks. BLINDER consistently outperforms or matches these baselines when providing descriptions for the flan-T5 actor that it was trained with. BLINDER also generalizes well to the much larger GPT3.5 Turbo actor, despite being trained on data from the T5 actor. The GPT3.5 Turbo actor with BLINDER state descriptions outperforms both baselines on three of five tasks.

BLINDER also performs competitively against our manual baseline that is engineered to only include task relevant information in the state description. Figure 3 shows BLINDER outperforming the manual baseline in three of five novel tasks. Notably these are the tasks that require multi-step planning by first picking up the target item and second using it to cross the lava. We hypothesize that BLINDER does better on these tasks by learning that different state features are relevant at different points in a trajectory.

Overall, BLINDER demonstrates that it can learn to generalize state descriptions for new tasks of increasing complexity that involve entities not seen during training. It is also able to generate state descriptions that are intuitive enough to generalize to new LLM actors. This allows our method to be trained using smaller local models and then deployed with large black-box models.

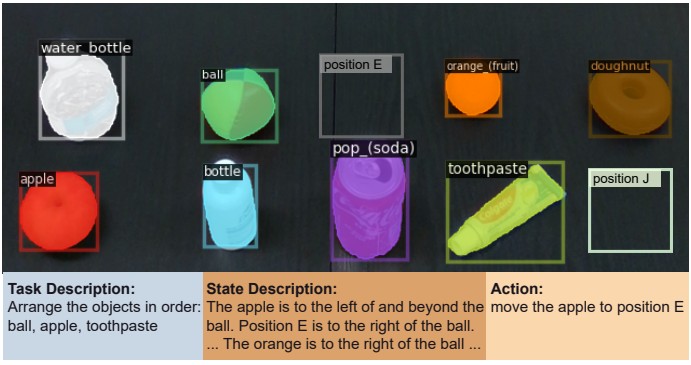

Figure 5: Segmented image of the robotic object arrangement task with example task and state descriptions and action. Objects are labeled and localized, relationships between objects are listed as state features, and an LLM actor chooses an object to move to an empty position.

## 6.2 EFFECTIVE CONTEXT USE

Most LLM actors have a limited input size, or context length, with which they are effective. Also, longer contexts result directly in more expensive compute. Success often requires effective use of context length, and with shorter state descriptions we can apply the freed context to improving the quality of our LLM actor.

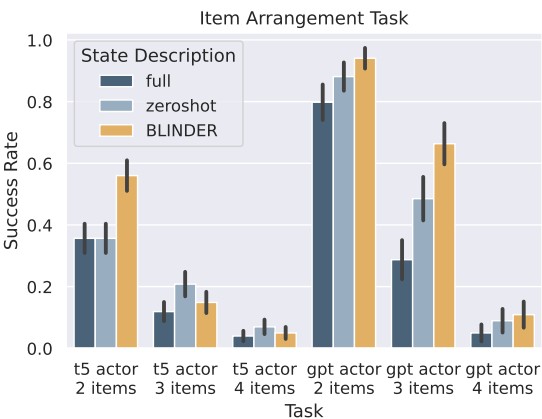

Figure 6: Success rates on our robotic object arrangement task when using BLINDER vs. baselines for state descriptions. The actor is tasked with arranging two, three, or four items. The GPT3.5 Turbo Actor once again outperforms the T5 actor, and BLINDER continues to generalize well despite being trained with the T5 actor.

Figure 4 shows how BLINDER drastically reduces context length by a factor of 6x and allows more effective use of our fewshot LLM actor's context. Shorter context lengths translate directly to cheaper inference time and costs. Also, with shorter contexts, we can include more fewshot examples and improve LLM actor performance.

## 7 ROBOTIC EXPERIMENTS

We also evaluate BLINDER on a real-world robotic object arrangement task. In this task, several objects must be rearranged on a table in order from left to right. The table is divided into a 2x5 grid of 10 possible positions where individual objects may reside. Various distractor objects are also placed in this grid to complicate the state space. BLINDER is trained using 25 expert demonstrations arranging two or three items. We evaluate on arranging two, three, or four items from a set of items not seen during training. See Appendix D for more details on the robitic task setup.

The robot environment uses an object segmentation model to map visual observations to natural language labels as shown in Figure 5. Detected object labels and coordinates are used to generate a set of text features $\hat{\Omega}$ describing spatial relationships between each object in the scene. For example, a table with an apple, banana, and orange arranged from left to right would result in the features *the orange is to the right of the apple*, *the banana is to the left of the orange*, etc. The locations of any empty grid positions are also included in $\hat{\Omega}$.

In this environment, the actions available to the LLM actor consist of specifying both which object to move and which unoccupied grid position to move it to. Our setup has ten possible grid positions and seven to nine objects, creating 90 state features in $\hat{\Omega}$, and 9-21 admissible actions at each step. Note that state descriptions for the robotic arrangement tasks are 3x longer on average than the NetHack task (see Tables 6 and 7). After the LLM actor selects an action, the robot then executes the low-level control to move the specified object to its new grid position.

Figure 6 shows the benefit of using BLINDER to select state descriptions for this task. Like our NetHack task, we compare to "full" and "zeroshot" baselines with zeroshot T5 and fewshot GPT3.5 Turbo actors. As with before, we use BLINDER trained with the T5 actor to generate descriptions for the GPT3.5 Turbo actor, and BLINDER successfully generalizes between actors. In fact, we found that the T5 actor struggled across the board to perform well at this task, likely due to the spatial and sequential reasoning required. However, consistent with the results from our NetHack experiments, BLINDER successfully selects state descriptions that improve actor performance further supporting the general applicability of our method.

## 8 DISCUSSION & CONCLUSION

Most prior work that evaluates the potential for using LLMs as actors in sequential decision making tasks ignore the issue of defining state descriptions. Instead they use exhaustive non-optimal state descriptions (Valmeekam et al., 2022) or hand-engineered task-specific state descriptions (Huang et al., 2022b; Liang et al., 2022; Singh et al., 2022). However, for LLM actors to be deployed in real world settings as general problem solvers, they will need to the ability to filter state features for novel tasks automatically. Specifically, in environments where the number of state spaces can be exponentially high. To this end, we introduce Brief Language INputs for DEcision-making Responses (BLINDER).

We not only show that selecting learned state descriptions improves actor performance by reducing distracting information, but we also demonstrate how the decreased context length can provide additional benefits. We show that the tasks can be performed with a very small model. In our experiments, our methods reduce the needed context length for decision making by a factor of 6x, which can translate directly to decreased inference costs. Alternatively, the additional context space can be applied to including more in-context examples and further improving performance of fewshot actors.

We believe that using LLM output to learn intuitive, textual inputs can be a powerful method for improving prompting and context efficiency. While our method focuses on selecting a prompt from an existing set of textual features, we encourage additional research that extends this idea to free-form summaries or feature abstractions. These ideas are also not limited to sequential decision making and should be researched in the context of other problem domains as well.

LLMs are being applied to a greater number of problems that require increasingly large context lengths. We believe that learning task-conditioned LLM input is a powerful method for improving performance and decreasing inference costs. We hope to encourage continued research into using LLM contexts more efficiently through methods such as pruning, summarization, and retrieval.

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

APPENDIX

## A    BLINDER DETAILS

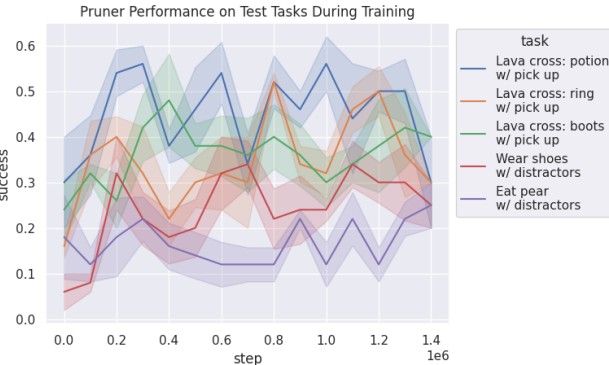

Figure 7: Performance on NetHack domain tasks with the T5 actor over the course of BLINDER training.

| Hyperparameter | Value |
|---|---|
| $\gamma$ | 1 |
| lr | 1e-6 |
| batch size | 4 |
| KL regularization coefficient | 1 |
| max state description length (nethack) | 10 |
| max state description length (robot) | 5 |

Table 2: BLINDER training hyperparameters

Table 2 shows hyperparameters used during BLINDER training. We finetune a `flan-t5-large` model with a value head after the final hidden state in the decoder. We prompt the model with the following instructions to take advantage of flan-T5's instruction finetuning:

```
Describe the relevant information from the game state for
the current task. Your current task is to [task description].
```

We finetune on monte carlo value estimates from the datasets described in Section 4.2. We also upsample instances from environment steps with less common actions. Figure 7 shows LLM actor success rate on NetHack test tasks throughout training.

## B    ACTOR DETAILS

### B.1    T5 ACTOR

We prompt a pretrained `flan-t5-xl` model with the below prompts for use as an LLM actor:

**NetHack Domain:**

```
You are playing the rogue-like game NetHack. Your task is to
[task description]. You can move north, south, east, west,
northeast, southeast, southwest, or northwest. You can attack
monsters adjacent to you, pick up items under you, zap wands,
eat food, wear armor, use keys, drink potions, and put on
rings. [state description]
You choose to:
```

**Robot Domain:**

```
You are controlling a helpful household robot. Your task is to
[task description]. You can move items from their current
positions to empty positions indicated by their cooresponding
letter. [state description]
You choose to:
```

At each task step, we compute the likelihood of the admissible actions, normalize the probabilities using the softmax function, and then sample an action to execute in the environment.

### B.2 GPT ACTOR

Our fewshot actor is a `gpt3.5-turbo-0301` model prompted with the below system messages:

**NetHack Domain:**

```
You are playing the rogue-like game NetHack. You can move north,
south, east, west, northeast, southeast, southwest, or
northwest. You can attack monsters adjacent to you, pick up
items under you, zap wands, eat food, wear armor, use keys,
drink potions, and put on rings.
```

**Robot Domain:**

```
You are controlling a helpful household robot. You can move
items from their current positions to empty positions indicated
by their cooresponding letter.
```

We provide fewshot examples and prompt the actor with the below prompt:

```
Your task is to [task description].

Game Description:
[state description]

Choose the best action:
[list of admissible actions]
```

The model generates an admissible action that is executed in the environment.

## C EXAMPLE TRAJECTORIES

Table 3 shows a successful NetHack trajectory for the "Lava Crossing: Potion" task, and tables 4 and 5 show successful trajectories for the robot arrangement task with three and four target items. All examples were collected with the fewshot GPT3.5 Turbo actor.

| Time | Image | Selected State Description | Action |
|---|---|---|---|
| t=0 |  | You see a effervescent potion very near south southwest. You see a stairs down near east southeast. You see a lava near south southeast. You have a +2 sling (alternate weapon; not wielded). | move southwest |
| t=1 |  | You see a effervescent potion very near south southwest. You see a stairs down near east southeast. You see a lava near south southeast. You have a +2 sling (You see a effervescent potion adjacent south. You have a +2 sling (alternate weapon; not wielded). Score: 0. You have an uncursed +0 leather armor (being worn). You have 16 uncursed flint stones (in quiver pouch). | move south |
| t=2 |  | You see here an effervescent potion. You have a +2 sling (alternate weapon; not wielded). Score: 0. Depth: 1. | pick up an effervescent potion |
| t=3 |  | f - an effervescent potion. You have a +2 sling (alternate weapon; not wielded). Score: 0. Encumbrance: Unencumbered. | drink an effervescent potion |
| t=4 |  | Dexterity: 13. You have 16 uncursed flint stones (in quiver pouch). You have 25 uncursed rocks. | move east |
| t=5 |  | Dexterity: 13. You have 16 uncursed flint stones (in quiver pouch). You have 25 uncursed rocks. | move east |
| t=6 |  | Dexterity: 13. You have 16 uncursed flint stones (in quiver pouch). You have 25 uncursed rocks. | move east |
| t=7 |  | Dexterity: 13. You have 16 uncursed flint stones (in quiver pouch). You have 25 uncursed rocks. | move east |
| t=8 |  | You see a stairs down adjacent northeast. You have 16 uncursed flint stones (in quiver pouch). You have 25 uncursed rocks. | move northeast |
| t=9 |  | - | - |

Table 3: Here, our fewshot actor is prompted with "Your task is to pick up and drink the potion and navigate to the stairs down.". The table shows a NetHack visualization alongside selected state descriptions and the action selected by the LLM actor.

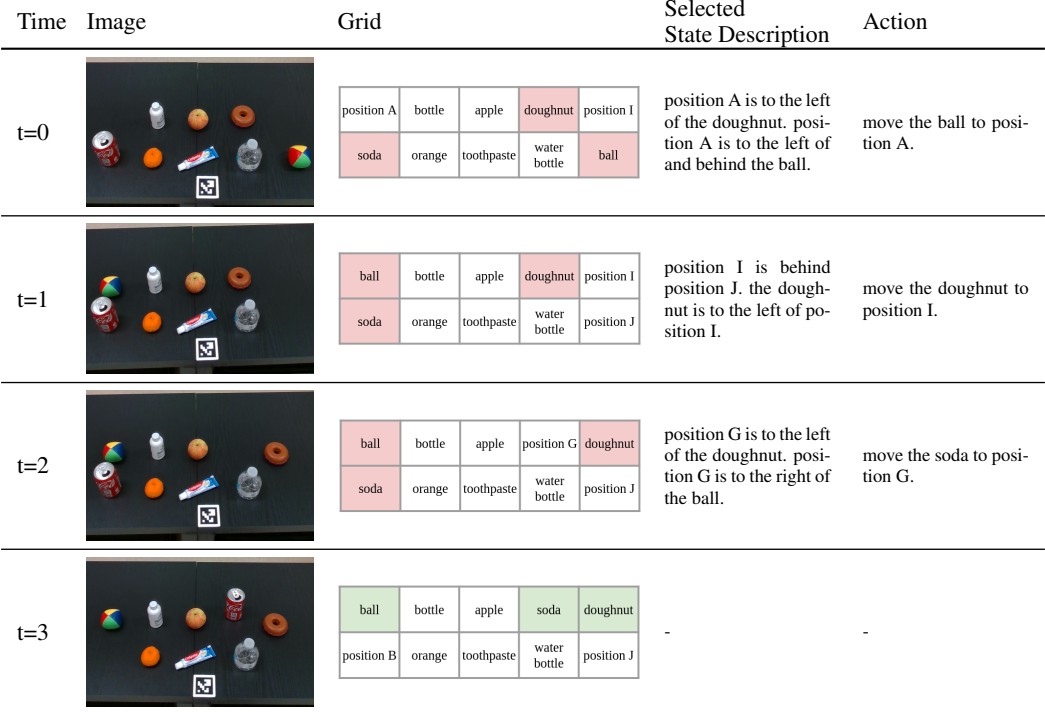

| Time | Image | Grid | Selected State Description | Action |
|------|-------|------|---------------------------|--------|
| t=0 | | position A / bottle / apple / doughnut / position I ; soda / orange / toothpaste / water bottle / ball | position A is to the left of the doughnut. position A is to the left of and behind the ball. | move the ball to position A. |
| t=1 | | ball / bottle / apple / doughnut / position I ; soda / orange / toothpaste / water bottle / position J | position I is behind position J. the doughnut is to the left of position I. | move the doughnut to position I. |
| t=2 | | ball / bottle / apple / position G / doughnut ; soda / orange / toothpaste / water bottle / position J | position G is to the left of the doughnut. position G is to the right of the ball. | move the soda to position G. |
| t=3 | | ball / bottle / apple / soda / doughnut ; position B / orange / toothpaste / water bottle / position J | - | - |

Table 4: Successful trajectory for the task "Arrange the objects in the order: **ball, soda, doughnut.**" The items relevant to the task are highlighted – green color denotes correct alignment and red means incorrect alignment.

| Time | Image | Grid | Selected State Description | Action |
|------|-------|------|---------------------------|--------|
| t=0 | | apple / position C / bottle / soda / water bottle ; ball / toothpaste / position F / orange / doughnut | the bottle is to the right of position C. position C is behind the toothpaste. position C is to the right of and behind the ball. | move the ball to position C. |
| t=1 | | apple / ball / bottle / soda / water bottle ; position B / toothpaste / position F / orange / doughnut | position B is to the left of the toothpaste. the ball is behind the toothpaste. position F is to the right of position B. the orange is to the right of position B. the soda is to the right of the ball. | move the toothpaste to position B. |
| t=2 | | apple / ball / bottle / soda / water bottle ; toothpaste / position D / position F / orange / doughnut | - | - |

Table 5: Successful trajectory for the task "Arrange the objects in the order: **toothpaste, ball, bottle, soda.**" Notably, BLINDER can be generalized to arranging an unseen number of items.

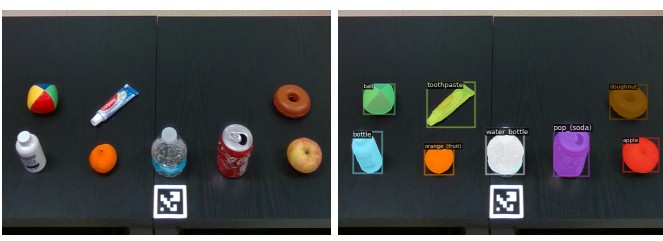

(a) RGB image (left) and corresponding segmentation result (right).

| position A | position C | position E | position G | position I |
|---|---|---|---|---|
| position B | position D | position F | position H | position J |

(b) Labels assigned to empty cells.

| ball | toothpaste | position E | position G | doughnut |
|---|---|---|---|---|
| bottle | orange | water bottle | soda | apple |

(c) The grid layout corresponding to RGB image on the left.

Figure 8

# D   ROBOT TASK DETAILS

## D.1   ROBOT DETAILS

For the robotics experiment, we use the Stretch RE2 (Kemp et al., 2022) from Hello Robot Inc. Stretch is a lightweight, low-cost mobile manipulator equipped with a variety of sensors, including an RGB-D camera and a 2D LiDAR (see Figure 9). The Stretch uses an Intel RealSense D435i camera to collect RGB and depth images of the table and we use both the onboard RP-LiDAR A1 and an added HTC Vive motion tracker for position estimates.

## D.2   ENVIRONMENT DETAILS

The goal of the robot manipulation task is to rearrange several objects on a table in a target order from left to right. Objects may each occupy and be placed in predefined locations in a 2-row by 5-column grid on the table. Target object arrangements are defined by the horizontal order of each item, and an object's final row location does not affect task success. A trial is successful if the robot arranges the objects in a specified order in 10 actions or less.

To investigate the generalization ability of BLINDER, we test on held-out items not seen during training. Additionally, while agents are tasked with arranging only two or three items during training, they are evaluated on arranging two, three, and four items at test time.

### D.2.1   OBSERVATION SPACE

Using LLMs as high-level planners requires natural language input, however object locations are perceived by the Stretch robot with camera inputs. We apply a pipeline to parse image observations to a canonical state representation and then to natural language state descriptions.

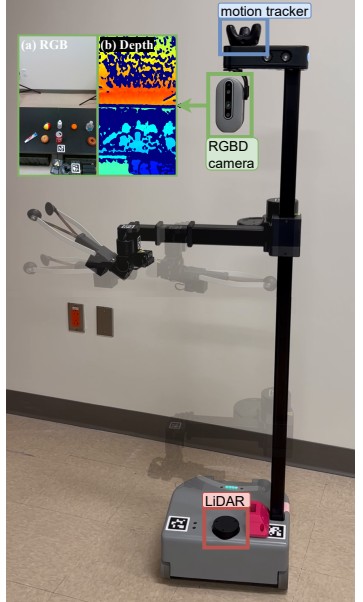

Figure 9: Stretch RE2.

Given an RGB image of the table, we detect table objects with an off-the-shelf semantic segmentation mask and determine their 3D location using a corresponding depth image. The locations of grid cells are implicitly defined relative to the table. The canonical state we extract describes which objects are in which grid cells. For segmentation, we use a `Mask R-CNN` (He et al., 2017) model with a ResNet-101-FPN backbone pre-trained on LVIS (Gupta et al., 2019) using the Detectron2 library (Wu et al., 2019). Figure 8a shows an example of an RGB image and its corresponding segmentation result. Figure 8c shows the parsed grid state corresponding to the image in Figure 8a.

Next, we construct a natural language state description from the current grid layout by listing spatial relationships between each grid cell and every other grid cell. Spatial relationships include left, right, behind, and beyond. For grid cells that are populated, the residing object labels are used as identifiers for the grid cells. Table 7 shows an example of a grid state being converted to a full natural language state description.

### D.2.2 ACTION SPACE

The high-level action defined for object arrangement follows the format "*move object_name to empty_position_name*". In our experiments, the Stretch robot achieves this by using 3D object coordinates obtained from segmented RGB-D camera images in combination with IKPy (Manceron, 2022), a python library for inverse kinematics.

| NetHack | Full State Description |
|---|---|
| 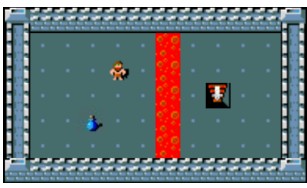 | "You have a +1 club (weapon in hand). You have a +2 sling (alternate weapon; not wielded). You have 16 uncursed flint stones (in quiver pouch). You have 25 uncursed rocks. You have an uncursed +0 leather armor (being worn). Strength: 22/19. Dexterity: 13. Constitution: 15. Intelligence: 9. Wisdom: 10. Charisma: 6. Depth: 1. Gold: 0. HP: 16/16. Energy: 2/2. AC: 8. XP: 1/0. Time: 1. Position: 36—9. Hunger: Not Hungry. Monster Level: 0. Encumbrance: Unencumbered. Dungeon Number: 0. Level Number: 1. Score: 0. Alignment: Neutral. Condition: None. You see a vertical wall far east. You see a stairs down near east southeast. You see a horizontal wall near southeast and south. You see a lava near south southeast. You see a southwest corner near southwest. You see a vertical wall near west. You see a horizontal wall very near north, northeast, and northwest. You see a lava very near east northeast, east, east southeast, and southeast. You see a effervescent potion very near south southwest. Hello Agent, welcome to NetHack! You are a neutral human Caveman" |

Table 6: The "Lava Cross: Potion" NetHack task observation and full state description.

| Grid | Full State Description |
|---|---|
| | "position E is behind the water bottle. position E is to the left of and behind the apple. position E is to the left of and behind the soda. position E is to the left of position G. position E is to the left of the doughnut. position E is to the right of and behind the bottle. position E is to the right of and behind the orange. position E is to the right of the ball. position E is to the right of the toothpaste. position G is behind the soda. position G is to the left of and behind the apple. position G is to the left of the doughnut. position G is to the right of and behind the bottle. position G is to the right of and behind the orange. position G is to the right of and behind the water bottle. position G is to the right of position E. position G is to the right of the ball. position G is to the right of the toothpaste. the apple is beyond the doughnut. the apple is to the right of and beyond position E. the apple is to the right of and beyond position G. the apple is to the right of and beyond the ball. the apple is to the right of and beyond the toothpaste. the apple is to the right of the bottle. the apple is to the right of the orange. the apple is to the right of the soda. the apple is to the right of the water bottle. the ball is behind the bottle. the ball is to the left of and behind the apple. the ball is to the left of and behind the orange. the ball is to the left of and behind the soda. the ball is to the left of and behind the water bottle. the ball is to the left of position E. the ball is to the left of position G. the ball is to the left of the doughnut. the ball is to the left of the toothpaste. the bottle is beyond the ball. the bottle is to the left of and beyond position E. the bottle is to the left of and beyond position G. the bottle is to the left of and beyond the doughnut. the bottle is to the left of and beyond the toothpaste. the bottle is to the left of the apple. the bottle is to the left of the orange. the bottle is to the left of the soda. the bottle is to the left of the water bottle. the doughnut is behind the apple. the doughnut is to the right of and behind the bottle. the doughnut is to the right of and behind the orange. the doughnut is to the right of and behind the soda. the doughnut is to the right of and behind the water bottle. the doughnut is to the right of position E. the doughnut is to the right of position G. the doughnut is to the right of the ball. the doughnut is to the right of the toothpaste. the orange is beyond the toothpaste. the orange is to the left of and beyond position E. the orange is to the left of and beyond position G. the orange is to the left of and beyond the doughnut. the orange is to the left of the apple. the orange is to the left of the soda. the orange is to the left of the water bottle. the orange is to the right of and beyond the ball. the orange is to the right of the bottle. the soda is beyond position G. the soda is to the left of and beyond the doughnut. the soda is to the left of the apple. the soda is to the right of and beyond position E. the soda is to the right of and beyond the ball. the soda is to the right of and beyond the toothpaste. the soda is to the right of the bottle. the soda is to the right of the orange. the soda is to the right of the water bottle. the toothpaste is behind the orange. the toothpaste is to the left of and behind the apple. the toothpaste is to the left of and behind the soda. the toothpaste is to the left of and behind the water bottle. the toothpaste is to the left of position E. the toothpaste is to the left of position G. the toothpaste is to the left of the doughnut. the toothpaste is to the right of and behind the bottle. the toothpaste is to the right of the ball. the water bottle is beyond position E. the water bottle is to the left of and beyond position G. the water bottle is to the left of and beyond the doughnut. the water bottle is to the left of the apple. the water bottle is to the left of the soda. the water bottle is to the right of and beyond the ball. the water bottle is to the right of and beyond the toothpaste. the water bottle is to the right of the bottle. the water bottle is to the right of the orange" |

The grid within the table shows:

| ball | toothpaste | position E | position G | doughnut |
|---|---|---|---|---|
| bottle | orange | water bottle | soda | apple |

Table 7: Robot arrangement task observation and full state description.

