# OpenReview forum: "Selective Perception: Learning Concise State Descriptions for Language Model Actors"
_ICLR.cc/2024/Conference — ICLR 2024 Conference Withdrawn Submission_

### Official Review · Reviewer_rLTn · 2023-10-31

**Soundness:** 2 fair
**Presentation:** 2 fair
**Contribution:** 2 fair
**Rating:** 5
**Confidence:** 4

**Summary:**

The paper proposes BLINDER, a module that learns to summarize task-relevant state features for the LLM actor. The primary motivation is that distractions in the full state description lead to bad performance of the LLM actor. The training pipeline aligns well with the MDP formulation, and the authors provide experiments to support the efficacy of BLINDER.

**Strengths:**

1. I fully agree with the motivation behind the work. Obtaining a succinct and informative state representation for LLM-based task planning is crucial since state representation is a major hurdle in applying LLM to task planning. However, VLM might significantly alleviate this situation.

2. In the work, the LLM actor is actually considered part of the reward function for BLINDER. As a result, the entire training pipeline aligns well with the MDP formulation, and the formulation is clear.

**Weaknesses:**

1. I believe a more robust baseline should be compared. Considering that the training of BLINDER utilizes privileged information, such as expert demonstrations, comparing it to zero-shot summarization might not provide a fair comparison. Moreover, with the evolving in-context learning capabilities of LLM, it's important to compare its performance with GPT-4's state feature extraction to understand the technical merits of BLINDER.

2. The experimental section would benefit from a clearer presentation. For instance, the terms 'zero-shot' and 'few-shots' are somewhat confusing. Some instances of 'zero-shot' referred to zero-shot summarization, while 'few-shot' referred to a few-shot LLM actor. The NetHack experiment is presented only in textual description, making it difficult for readers to visualize the experimental setup.

3. The paper would benefit from a more in-depth analysis. I would like to see some evidence, based on the experimental results, regarding the inherent limitations of LLM that necessitate an additional component like BLINDER. 1) It fails to utilize the relevant information amidst all the distractions, even with GPT-4, much like its struggles with simple 4-digit algorithms. 2) It cannot extract relevant information from the distractions, even with GPT-4 and despite examples and prompting. This would help underscore the significance of BLINDER at this moment.

**Update after rebuttal:**

I appreciate the author's response and clarification. Here are my thoughts after reading the rebuttal:

1. The improved performance after using GPT-4 isn't entirely unexpected, given the improved capabilities of a more advanced language model. While creating more challenging tasks is possible, I believe the bottleneck may mainly lie with the LLM agent rather than the state description. However, I agree that minimizing redundant state descriptions is valuable, particularly in aiding human understanding of the LLM’s reasoning process and in identifying key information for the model. Unfortunately, the manuscript lacks an analysis of this aspect.

2. I understand why the authors chose not to separate the state description from the LLM agent, considering BLINDER is tailored for a specific language model. Nonetheless, I believe there's merit in doing so. Whether certain states are relevant or not should be consistent across different language models. Since most language models are somewhat 'black boxes' and can be seen as repositories of commonsense knowledge, the internal reasoning mechanisms are not fully understood. Decoupling could potentially offer deeper insights into BLINDER, such as why it generates irrelevant information.

While BLINDER has potential as a general 'plug-in' for LLM agents, a more thorough analysis and deeper insights would enhance its value.

**Questions:**

1. In Figure 2, it says GPT-3; I guess that should be GPT-3.5?
2. In the reward function, how does one differentiate whether the low reward is due to an incorrect state description or the inability of the LLM to reason?
3. Does the order of state features affect performance?
4. How will BLINDER generalize to different state features without additional training?
5. Is there any reason for BLINDER to generate hallucinated state descriptions? This is a bit concerning.

---

> ### Author Response · Authors · 2023-11-20
>
> Thank you for your review. Unfortunately, we did not yet have access to GPT4 when originally running these experiments. We have since run them with GPT4 and it solves our tasks with close to 100% accuracy with and without BLINDER. However, BLINDER still decreases the state description length by an average of 83% while maintaining the high success rate of GPT4. We leave it up to future work to propose more complex tasks for challenging GPT4.
> We hope that the answers below clarify any doubts you have about our paper.
>
> 1. Yes, figure 2 is referring to gpt3.5-turbo, as mentioned in the text. We are correcting this misleading label.
> 2. We do not believe the distinction between low action likelihood due to the state description vs. LLM actor is necessary. Either way, we try to find the state description that best optimizes the action likelihood.
> 3. In our tasks the order of state features does not affect performance significantly. However, BLINDER's training setup is equipped to learn the best order when necessary.
> 4. BLINDER does generalize to different state features. All of our results are done on held out tasks that were not seen while training BLINDER. These test tasks have a different set of possible state features.
> 5. No. Unlike generative summarization methods, BLINDER is constrained to only include features found in the original state description.

---

### Official Review · Reviewer_Xdzb · 2023-10-31

**Soundness:** 2 fair
**Presentation:** 3 good
**Contribution:** 2 fair
**Rating:** 3
**Confidence:** 5

**Summary:**

The manuscript proposes a method for learning a value function for obtaining better state descriptions for LLM-based actors in interactive games and tabletop object rearrangement tasks.

**Strengths:**

The paper is well-written

The paper addresses an important problem is utilizing LLMs for sequential task-execution

**Weaknesses:**

Section 1 — The manuscript states that the "the system should learn to automatically generate task-conditioned state descriptions". The manuscript should discuss here about the proposed mechanisms that allow the learning process remain both general and task-customized enough, to have the best of both worlds. It would be detrimental to the novelty if the generalization of the feature selection process is too dependent on the set of tasks it is predicated on. Discuss in the main content how the tasks and their expert trajectories were chosen.

Section 1 — The manuscript states that "[u]nlike prior work, BLINDER automatically constructs state descriptions without requiring task- specific prompt engineering." This is not strong a novelty, as BLINDER requires a carefully-chosen set of set of archetypal tasks.

Section 1 — "Given a small set of expert task demonstrations ..." — Certainly, if the tasks are seen, there would be little novelty to the proposed method. However, if the tasks are unseen during training, how can we say that the value function is trustworthy? Would there not be a distribution shift in value (which is conditioned on the expert set of tasks), just as there is in the set of tasks itself? The inference we can draw seems only to be that the training tasks are not significantly different from the test tasks.

Section 2.1-2.2 — It is not sufficient to merely survey the related work. In each subsection, which compares the proposed work with prior art (according to a specified theme or dimension), the manuscript should identify salient weaknesses and concisely specify how the proposed approach alleviates these concerns. For the rebuttal, please provide these statements, for each subsection.

Section 2.1 — The manuscript is missing discussion about how its proposed approach alleviates the grounding problem, which it identified as the prevailing weakness of the prior art in the ‘Language for State Descriptions’ dimension.

Section 2.2 — Same issue as above: missing discussion of how the proposed approach alleviates the weaknesses

Section 2.3 — "[the] approach [in Zhang et al., (2023)] differs significantly from our own and focuses on static NLP tasks rather than sequential tasks." This is not a sufficient basis for discrimination, as the approach proposed by the prior art, Zhang et al., (2023) can be very easily applied to sequential problems, since each decision-making step is, itself, a static NLP task. In the present manuscript’s problem formulation, nothing about the state description process depends on the surrounding agent’s action-generation.

Section 3 — According to this formulation, the state features seem only additive. Are there situations in which the reverse formulation (i.e., starting with omega-hat and iteratively removing irrelevant features) may be better? Would the same value function be learned? Why did you chose this formulation, or why does it not matter?

Section 3 — Does the “size” (or “capacity”) of the state representation only monotonically increase? What does this look like for long-horizon tasks?

Section 4.1 — The manuscript states "We use π to construct a state description x out of state features ω. Starting with x0 = ∅, π iteritively [sic] adds ωt∗ to xt as long as it increases in value and there are still state features to add (see Algorithm 1)." Does the policy learn an ordering bias? The manuscript should provide an “ablation” in which the ordering of the state elements is perturbed, at the end of each feature selection process.

Section 4.2 — the manuscript states "we reward BLINDER to maximize the likelihood that xf elicits the same action from the LLM actor as the target action a∗ from the expert trajectory." This carries an implicit assumption that the expert trajectories are somehow representative archetypes for test-time execution environments. If they aren’t representatives, the reward structure could be severely biased towards the training environment; if they are representative, the test-time execution environment would then not be so different from the training environment and the claim of unseen generalisation suffers (and the empirical performance improvement would be insignificant).

Table 1 (Caption) — The manuscript states "Although BLINDER sometimes includes unnecessary information, any extra included state features do not distract from the current task." How should we know? The manuscript should provide evidence that removal of this unnecessary information has no effect on performance.

Section 4.3 — The manuscript states "While we refer to x as a set, the text input to Vθ is ordered using x’s insertion order. " Ordering bias? See comment above

Section 5 — The manuscript states "In each of our experiments, BLINDER is trained on a set of training tasks and is provided five expert trajectories for each task." Missing discussion about how expensive it should be expected to be to provide five expert trajectories for each task, if the proposed method were applied to different settings. Does the required number of expert trajectories need to change as the number of tasks changes? What is the scaling factor for the number of demonstrations needed for an arbitrary number of (diverse) tasks?

Section 5.1 — Lack of fair comparison / strong baseline. The manuscript states "On average, BLINDER removes 83% of the original state features." Unfortunately, by itself, this number is meaningless. The manuscript should adapt an alternative approach as a strong baseline, e.g., from Zhang et al., (2023), in order to produce fair comparisons. Then, the manuscript could discuss how much distracting information a strong baseline or ablation removes, here, compared to the proposed approach

Section 6 (Intro) — An entire subsection should be dedicated to describing why just these selected tasks were chosen for training and testing. The current discussion is insufficient and leaves open the possibility that these tasks were selected to favour the proposed method. Why not evaluate on a much larger set of tasks? Why not evaluate on all the tasks? Why not organise different task splits, e.g., where the test variants are completely different from the training variants (rather than selecting test tasks that only differ by whether an object starts in the player’s inventory or not)?

Figure 6 (Caption) — It seems that a large-capacity model is needed for leveraging the state description produced by BLINDER.

Section 7 — Lack of fair comparison / strong baseline. Missing a fair comparison with a strong baseline that attempts test-time prompt tuning.

**Questions:**

N/A — see weaknesses above.

---

> ### Author Response · Authors · 2023-11-20
>
> Thank you very much for your thorough feedback. Hopefully, we can clarify a few points. The point of the paper is to learn a small feature selection model that can be paired with a larger actor to significantly decrease the length and cost of prompts while also improving performance. It is expected that this smaller feature selection model would generalize in domain to new state features, but may not perform well on out-of-domain tasks. Many previous works attempt to improve LLM actor grounding by including exhaustive descriptions of the state. BLINDER alleviates the issue of distracting information and expensive/long prompts caused by naively providing exhaustive information to the actor.
>
> Other notes:
> - Zhang et al. (2023) is concurrent work to ours also under review at ICLR this year.
> - The benefit of iteratively adding to the state description rather than pruning it lies in the fact that you never need to have the entire state description in context at one time, thus reducing compute.
> - The length of the downstream task has no effect on the length of state descriptions for individual steps.
> - As you mention, BLINDER is equipped to learn a specific order of state features. In our experiments the order of state features does not affect performance significantly. However, BLINDER's training setup is equipped to learn the best order when necessary.
> Hopefully these points clarified any misconceptions you had and can improve your rating of our work.

---

### Official Review · Reviewer_7C4S · 2023-11-01

**Soundness:** 4 excellent
**Presentation:** 3 good
**Contribution:** 3 good
**Rating:** 6
**Confidence:** 4

**Summary:**

The paper proposes BLINDER, a method for automatically selecting task-relevant text features for input to an LLM actor. On NetHack and a robot manipulation task, BLINDER decreases the length of the input text while improving performance relative to the full-text input. BLINDER also learns intuitive text selection and zero-shot generalizes to larger, more powerful LLMs.

**Strengths:**

- Experiments demonstrate that BLINDER improves the success rate over providing a full, zero-shot LLM generated and manually engineered state description (Fig 2).
- BLINDER is evaluated on two diverse domains: Nethack and a robotic manipulation task.
- BLINDER learns generalizable state descriptions. The method is able to generalize to variants of the Nethack training tasks and is able to transfer the descriptions zero-shot to a different LLM.
- The paper includes qualitative examples of the selected state description and how it evolves throughout the episode.

**Weaknesses:**

- BLINDER can select relevant text features, but will this be important as LLM context lengths increase? NetHack has at most 40 features, and the robotic task has 90 features. A more powerful LLM, like GPT-4 could perform better from the full state description. Sec. 6.2 is perhaps intended to address this point, yet the section feels incomplete. It doesn't discuss the results in Fig. 4, and Fig. 4 is not discussed elsewhere in the text.
- BLINDER requires manually defining the set of grounded state features. Defining these features could be more difficult in more complex environments involving open-ended descriptions of environments. Instead, BLINDER will only work in environments where the state consists of a set of fixed state features.
- BLINDER requires a precollected dataset of experience to learn the value function. The paper collects this dataset with a random policy. How can this scale to more complex problems where this random policy would not collect meaningful data?

**Questions:**

- Why does BLINDER outperform the manual state descriptions?
- What prompt is used to zero-shot summarize the state description (zeroshot baseline in Fig 2)? Is it the same prompt as in 5.1? Why not use GPT-3.5 as the summarizer since it is a more powerful LLM than Flan-T5 (as demonstrated by Fig 2)?
- What do the error bars in Fig 2 represent? This is not described in the caption or elsewhere in the paper.
- Section 2.3 states that prior works that explore compressing input tokens for shorter context lengths don't meet the needs of sequential decision-making. Why is that the case, and why can't such methods be used in the same setting as BLINDER (at least in the case of Flan-T5 Actor)?
- What does Fig. 4 mean by "context length per action"? What exactly does the x-axis quantity refer to in Fig. 4?
- Table 1 separates state descriptions into relevant and irrelevant text with blue and red colors. But what does the black text in the state description from BLINDER correspond to?

---

> ### Author Response · Authors · 2023-11-20
>
> Thank you for your review! Although many recent LLMs have context lengths that can fit in more data than we use here, longer contexts mean higher compute costs and time. One major benefit of our method is that you can use smaller models to shorten the context for more expensive models. While we started by training BLINDER with data from a random policy, we also tried continuing to collect data using the current BLINDER policy. We didn't find any further improvement, but in some environments this may be necessary for good results.
>
> 1. Our manual baseline includes any features from the state description that mention an entity also mentioned in the task description. This is not meant to be an oracle but a simple hand engineered approach. BLINDER outperforms this approach because the manual state descriptions are not state dependent while BLINDER is.
> 2. We use the prompt from Appendix section A. We could have used gpt3.5-turbo as the summarizer and actor in Figure 2b. Our intention was to use a model in the same family as BLINDER for zeroshot summarization,  and we were not concerned about matching the actor.
> 3. The error bars are standard error of successes vs. failures from a single trained model.
> 4. While some autoprompt-like [1] methods from nlp may work in a sequential setting, one advantage of our work is that state descriptions remain interpretable and generalize between actors. The only work we are aware of that meets this criteria is TEMPERA which is concurrent to our work.
> [1]:Shin, Taylor, et al. "Autoprompt: Eliciting knowledge from language models with automatically generated prompts." arXiv preprint arXiv:2010.15980 (2020).
>
> 5. The x-axis in Figure 4 refers to the average context length of the LLM actor for each environment step.
> 6. The caption labels blue text as relevant and red text as inaccurate/distracting. The remaining black text is irrelevant but not distracting or inaccurate in any way.

---

### Official Review · Reviewer_6BNo · 2023-11-01

**Soundness:** 3 good
**Presentation:** 3 good
**Contribution:** 2 fair
**Rating:** 3
**Confidence:** 3

**Summary:**

The authors propose BLINDER, a method for selecting state descriptions for LLM actors in sequential decision making problems.  It does this by learning a value function for task-conditioned state descriptions that approximates the likelihood that the descriptions will lead to optimal actions.  BLINDER reduces context length and removes distracting information forming an optimal state description for the actor.

**Strengths:**

The paper is well written and clear.  Technical details are easy to follow and the motivation is clear.  Evaluations are provided on multiple applications and compared to a competing approach.

**Weaknesses:**

"We hypothesize that BLINDER does better on these tasks by learning that different state features are relevant at different points in a trajectory". Are there some qualitative examples that show this?  It would be interesting to see how different features are selected at different stages of the task.

In Figure 2, BLINDER is only achieving significant improvement in 3/5 tasks with the flan-T5 actor and 2/5 with the GPT3 actor.  In Figure 3, BLINDER shows significant improvement over manually specified descriptions in 3/5 tasks.  In Figure 6, BLINDER only shows significant improvement on 2/6 tasks.  These results are not very convincing given the simplicity of the zeroshot approach in comparison.  Can the authors better highlight the benefits of the approach and potential tradeoffs of the zeroshot approach?  I think the context length section touched upon it but it was not immediately clear why I would use the proposed approach over the simple LLM feature selection.

**Questions:**

see above

---

> ### Author Response · Authors · 2023-11-20
>
> Thank you for your review of our paper! As you pointed out BLINDER doesn't always beat baseline performance by a significant margin. However, it at least performs as well as baselines with a significantly shorter context length. We urge you to consider that in addition to providing comparable if not better performance, BLINDER always significantly reduces time and computing costs. Our paper explains that BLINDER removes 83% of state features, and as shown in Figure 4 this equates to a significant decrease in context length especially in fewshot settings. BLINDER is able to do so without ever needing the entire state description in context at once. Also, we are able to construct relevant state descriptions with much smaller model sizes using BLINDER compared to our zeroshot summarization baselines. We urge you to reconsider your rating in light of these points.
> Also, we have qualitative examples of BLINDER behavior in the Appendix Tables 3-5.

---

### Author Response · Authors · 2023-11-20

Dear Reviewers,

We are grateful for your constructive feedback and positive remarks on our paper. We appreciate Reviewer 1's acknowledgment of our evaluations on multiple applications and the comparison to competing approaches. Reviewer 2's recognition of BLINDER’s evaluation across diverse domains is highly encouraging. Additionally, Reviewer 3's comment on the paper addressing the significant problem of utilizing large language models (LLMs) for sequential task-execution is particularly motivating. These acknowledgments validate the efforts we have made in our research.

In the following sections, we aim to address the common questions and concerns raised by the reviewers. Our hope is that through this dialogue, we can provide further clarity and enhance the overall quality and impact of our work.

One of the major contributions of BLINDER is a method that trains a small feature selection model to lower the cost of and improve performance for a larger actor model. This reduction in context length can further be applied to environments where there are exponentially many features. We are working on improved figures that better emphasize the cost-benefit of reduced prompt lengths.

Another concern was our lack of comparison with previous work. All previous NLP approaches for learning LLM inputs focus on maximizing performance on a dataset by conditioning on a special set of learned uninterpretable tokens. These methods do not accomplish our goal of decreasing context lengths. While a method similar to Zhang et al. (2023) could be adapted to edit our state features into more concise descriptions, this was not within the scope of that paper. In addition, that work is concurrent with ours and also under review at this conference.

Our paper introduces a method for creating concise inputs for LLM actors that takes advantage of the independent nature of state features in embodied environments to construct state descriptions. We have the dual objective of improving downstream performance and decreasing context lengths. Our method is unique in its application and purpose because it is the first work that addresses the specific concerns of crafting model inputs for embodied ai. Also, compared to previous work in NLP that learns model inputs, our method prioritizes short contexts and natural inputs. We hope that the reviewers will consider increasing their ratings of our paper in consideration of these points.